# Linear dynamical neural population models through nonlinear embeddings

**Yuanjun Gao**[*1] , **Evan Archer**[*12], **Liam Paninski**[12], **John P. Cunningham**[12]
Department of Statistics[1] and Grossman Center[2]
Columbia University
New York, NY, United States
`yg2312@columbia.edu, evan@stat.columbia.edu,`
`liam@stat.columbia.edu, jpc2181@columbia.edu`

## Abstract

A body of recent work in modeling neural activity focuses on recovering low-dimensional latent features that capture the statistical structure of large-scale neural populations. Most such approaches have focused on linear generative models, where inference is computationally tractable. Here, we propose fLDS, a general class of nonlinear generative models that permits the firing rate of each neuron to vary as an arbitrary smooth function of a latent, linear dynamical state. This extra flexibility allows the model to capture a richer set of neural variability than a purely linear model, but retains an easily visualizable low-dimensional latent space. To fit this class of non-conjugate models we propose a variational inference scheme, along with a novel approximate posterior capable of capturing rich temporal correlations across time. We show that our techniques permit inference in a wide class of generative models.We also show in application to two neural datasets that, compared to state-of-the-art neural population models, fLDS captures a much larger proportion of neural variability with a small number of latent dimensions, providing superior predictive performance and interpretability.

## 1   Introduction

Until recently, neural data analysis techniques focused primarily upon the analysis of single neurons and small populations. However, new experimental techniques enable the simultaneous recording of ever-larger neural populations (at present, hundreds to tens of thousands of neurons). Access to these high-dimensional data has spurred a search for new statistical methods. One recent approach has focused on extracting latent, low-dimensional dynamical trajectories that describe the activity of an entire population [1, 2, 3]. The resulting models and techniques permit tractable analysis and visualization of high-dimensional neural data. Further, applications to motor cortex [4] and visual cortex [5, 6] suggest that the latent trajectories recovered by these methods can provide insight into underlying neural computations.

Previous work for inferring latent trajectories has considered models with a latent linear dynamics that couple with observations either linearly, or through a restricted nonlinearity [1, 3, 7]. When the true data generating process is nonlinear (for example, when neurons respond nonlinearly to a common, low-dimensional unobserved stimulus), the observation may lie in a low-dimensional nonlinear subspace that can not be captured using a mismatched observation model, hampering the ability of latent linear models to recover the low-dimensional structure from the data. Here, we propose fLDS, a new approach to inferring latent neural trajectories that generalizes several previously proposed methods. As in previous methods, we model a latent dynamical state with a

---

[*]These authors contributed equally.

linear dynamical system (LDS) prior. But, under our model, each neuron's spike rate is permitted to vary as an arbitrary smooth nonlinear function of the latent state. By permitting each cell to express its own, private non-linear response properties, our approach seeks to find a nonlinear embedding of a neural time series into a linear-dynamical state space.

To perform inference in this nonlinear model we adapt recent advances in variational inference [8, 9, 10]. Using a novel approximate posterior that is capable of capturing rich correlation structure in time, our techniques can be applied to a large class of latent-LDS models. We show that our variational inference approach, when applied to learn generative models that predominate in the neural data analysis literature, performs comparably to inference techniques designed for a specific model. More interestingly, we show in both simulation and application to two neural datasets that our fLDS modeling framework yields higher prediction performance with a more compact and informative latent representation, as compared to state-of-the-art neural population models.

## 2  Notation and overview of neural data

Neuronal signals take the form of temporally fast ($\sim 1$ ms) spikes that are typically modeled as discrete events. Although the spiking response of individual neurons has been the focus of intense research, modern experimental techniques make it possible to study the simultaneous activity of large numbers of neurons. In real data analysis, we usually discretize time into small bins of duration $\Delta t$ and represent the response of a population of $n$ neurons at time $t$ by a vector $\mathbf{x}_t$ of length $n$, whose $i^{th}$ entry represents number of spikes recorded from neuron $i$ in time bin $t$, where $i \in \{1, \ldots, n\}$, $t \in \{1, \ldots, T\}$. Additionally, because spike responses are variable even under identical experimental conditions, it is commonplace to record many repeated trials, $r \in \{1, \ldots, R\}$, of the same experiment.

Here, we denote $\mathbf{x}_{rt} = (x_{rt1}, ..., x_{rtn})^\top \in \mathbb{N}^n$ as spike counts of $n$ neurons for time $t$ and trial $r$. When the time index is suppressed, we refer to a data matrix $\mathbf{x}_r = (\mathbf{x}_{r1}, ..., \mathbf{x}_{rT}) \in \mathbb{N}^{T \times n}$. We also use $\mathbf{x} = (\mathbf{x}_1, ..., \mathbf{x}_R) \in \mathbb{N}^{T \times n \times R}$ to denote all the observations. We use analogous notation for other temporal variables; for instance $\mathbf{z}_r$ and $\mathbf{z}$.

## 3  Review of latent LDS neural population models

Latent factor models are popular tools in neural data analysis, where they are used to infer low-dimensional, time-evolving latent trajectories (or factors) $\mathbf{z}_{rt} \in \mathbb{R}^m, m \ll n$ that capture a large proportion of the variability present in a neural population recording. Many recent techniques follow this general approach, with distinct noise models [3], different priors on the latent factors [11, 12], extra model structure [13] and so on.

We focus upon one thread of this literature that takes its inspiration directly from the classical Kalman filter. Under this approach, the dynamics of a population of $n$ neurons are modulated by an unobserved, linear dynamical system (LDS) with an $m$-dimensional latent state $\mathbf{z}_{rt}$ that evolves according to,

$$\mathbf{z}_{r1} \sim \mathcal{N}(\mu_1, \mathbf{Q}_1) \tag{1}$$
$$\mathbf{z}_{r(t+1)}|\mathbf{z}_{rt} \sim \mathcal{N}(\mathbf{A}\mathbf{z}_{rt}, \mathbf{Q}), \tag{2}$$

where $\mathbf{A}$ is an $m \times m$ linear dynamics matrix, and the matrices $\mathbf{Q}_1$ and $\mathbf{Q}$ are the covariances of the initial states and Gaussian innovation noise, respectively. The spike count observation is then related to the latent state via an observation model,

$$x_{rti}|\mathbf{z}_{rt} \sim \mathcal{P}_\lambda \left( \lambda_{rti} = [f(\mathbf{z}_{rt})]_i \right). \tag{3}$$

where $[f(\mathbf{z}_{rt})]_i$ is the $i^{th}$ element of a deterministic "rate" function $f(\mathbf{z}_{rt}) : \mathbb{R}^m \to \mathbb{R}^n$, and $\mathcal{P}_\lambda(\lambda)$ is a noise model with parameter $\lambda$. Each choice among the ingredients $f$ and $\mathcal{P}_\lambda$ leads to a model with distinct characteristics. When $\mathcal{P}_\lambda$ is a Gaussian distribution with mean parameter $\lambda$ and linear rate function $f$, the model reduces to the classical Kalman filter. All operations in the Kalman filter are conjugate, and inference may be performed in closed form. However, any non-Gaussian noise model $\mathcal{P}_\lambda$ or nonlinear rate function $f$ breaks conjugacy and necessitates the use of approximate inference techniques. This is generally the case for neural models, where the discrete, positive nature of spikes suggests the use of discrete noise models with positive link[1, 3].

**Examples of latent LDS models for neural populations:** Existing LDS models usually impose strong assumptions on the rate function. When $\mathcal{P}_\lambda$ is chosen to be Poisson with $f(\mathbf{z}_{rt})$ to be the (element-wise) exponential of a linear transformation of $\mathbf{z}_{rt}$, we recover the *Poisson linear dynamical system* model (PLDS)[1],

$$x_{rti}|\mathbf{z}_{rt} \sim \text{Poisson}\left(\lambda_{rti} = \exp(c_i \mathbf{z}_{rt} + d_i)\right), \tag{4}$$

where $c_i$ is the $i$th row of the $n \times m$ observation matrix $\mathbf{C}$ and $d_i \in \mathbb{R}$ is the baseline firing rate of neuron $i$. With $\mathcal{P}_\lambda$ chosen to be a generalized count (GC) distribution and linear rate $f$, the model is called the *generalized count linear dynamical system* (GCLDS) [3],

$$x_{rti}|\mathbf{z}_t \sim \mathcal{GC}\left(\lambda_{rti} = c_i \mathbf{z}_{rt}, g_i(\cdot)\right). \tag{5}$$

where $\mathcal{GC}(\lambda, g(\cdot))$ is a distribution family parameterized by $\lambda \in \mathbb{R}$ and a function $g(\cdot) : \mathbb{N} \to \mathbb{R}$, distributed as,

$$p_{\mathcal{GC}}(k; \lambda, g(\cdot)) = \frac{\exp(\lambda k + g(k))}{k! M(\lambda, g(\cdot))}. \quad k \in \mathbb{N} \tag{6}$$

where $M(\lambda, g(\cdot)) = \sum_{k=0}^{\infty} \frac{\exp(\lambda k + g(k))}{k!}$ is the normalizing constant. The GC model can flexibly capture under- and over-dispersed count distributions.

# 4 Nonlinear latent variable models for neural populations

## 4.1 Generative Model: Linear dynamical system with nonlinear observation

We relax the linear assumptions of the previous LDS-based neural population models by incorporating a per-neuron rate function. We retain the latent LDS of eq. 1 and eq. 2, but select an observation model such that each neuron has a separate nonlinear dependence upon the latent variable,

$$x_{rti}|\mathbf{z}_{rt} \sim \mathcal{P}_\lambda\left(\lambda_{rti} = [f_\psi(\mathbf{z}_{rt})]_i\right), \tag{7}$$

where $\mathcal{P}_\lambda(\lambda)$ is a noise model with parameter $\lambda$; $f_\psi : \mathbb{R}^m \to \mathbb{R}^n$ is an arbitrary continuous function from the latent state into the spike rate; and $[f_\psi(\mathbf{z}_{rt})]_i$ is the $i^{th}$ element of $f_\psi(\mathbf{z}_{rt})$. In principle, the rate functions may be represented using any technique for function approximation. Here, we represent $f_\psi(\cdot)$ through a feed-forward neural network model. The parameters $\psi$ then amount to the weights and biases of all units across all layers. For the remainder of the text, we use $\theta$ to denote all generative model parameters: $\theta = (\mu_1, Q_1, A, Q, \psi)$. We refer to this class of models as fLDS.

To refer to an fLDS with a given noise model $\mathcal{P}_\lambda$, we prepend the noise model to the acronym. In the experiments, we will consider both PfLDS (taking $\mathcal{P}_\lambda$ to be Poisson) and GCfLDS (taking $\mathcal{P}_\lambda$ to be a generalized count distribution).

## 4.2 Model Fitting: Auto-encoding variational Bayes (AEVB)

Our goal is to learn the model parameters $\theta$ and to infer the posterior distribution over the latent variables $\mathbf{z}$. Ideally, we would perform maximum likelihood estimation on the parameters, $\hat{\theta} = \arg\max_\theta \log p_\theta(\mathbf{x}) = \arg\max_\theta \sum_{r=1}^{R} \int p_\theta(\mathbf{x}_r, \mathbf{z}_r) d\mathbf{z}_r$, and compute the posterior $p_{\hat{\theta}}(\mathbf{z}|\mathbf{x})$. However, under a fLDS neither the $p_\theta(\mathbf{z}|\mathbf{x})$ nor $p_\theta(\mathbf{x})$ are computationally tractable (both due to the noise model $\mathcal{P}_\lambda$ and the nonlinear observation model $f_\psi(\cdot)$). As a result, we pursue a stochastic variational inference approach to simultaneously learn parameters $\theta$ and infer the distribution of $\mathbf{z}$.

The strategy of variational inference is to approximate the intractable posterior distribution $p_\theta(\mathbf{z}|\mathbf{x})$ by a tractable distribution $q_\phi(\mathbf{z}|\mathbf{x})$, which carries its own parameters $\phi$.[2] With an approximate posterior[3] in hand, we learn both $p_\theta(\mathbf{z}, \mathbf{x})$ and $q_\phi(\mathbf{z}|\mathbf{x})$ simultanously by maximizing the *evidence lower bound* (ELBO) of the marginal log likelihood:

$$\log p_\theta(\mathbf{x}) \geq \mathcal{L}(\theta, \phi; \mathbf{x}) = \sum_{r=1}^{R} \mathcal{L}(\theta, \phi; \mathbf{x}_r) = \sum_{r=1}^{R} \mathbb{E}_{q_\phi(\mathbf{z}_r|\mathbf{x}_r)}\left[\log \frac{p_\theta(\mathbf{x}_r, \mathbf{z}_r)}{q_\phi(\mathbf{z}_r|\mathbf{x}_r)}\right]. \tag{8}$$

We optimize $\mathcal{L}(\theta, \phi; \mathbf{x})$ by stochastic gradient ascent, using a Monte Carlo estimate of the gradient $\nabla \mathcal{L}$. It is well-documented that Monte Carlo estimates of $\nabla \mathcal{L}$ are typically of very high variance, and strategies for variance reduction are an active area of research [14, 15].

Here, we take an auto-encoding variational Bayes (AEVB) approach [8, 9, 10] to estimate $\nabla \mathcal{L}$. In AEVB, we choose an easy-to-sample random variable $\epsilon \sim p(\epsilon)$ and sample $\mathbf{z}$ through a transformation of random sample $\epsilon$ parameterized by observations $\mathbf{x}$ and parameters $\phi$: $\mathbf{z} = h_\phi(\mathbf{x}, \epsilon)$ to get a rich set of variational distributions $q_\phi(\mathbf{z}|\mathbf{x})$. We then use the unbiased gradient estimator on minibatches consisting of a randomly selected single trials $\mathbf{x}_r$,

$$\nabla \mathcal{L}(\theta, \phi; \mathbf{x}) \approx R \nabla \mathcal{L}(\theta, \phi; \mathbf{x}_r) \tag{9}$$

$$\approx R \left[ \frac{1}{L} \sum_{l=1}^{L} \nabla \log p_\theta(\mathbf{x}_r, h_\phi(\mathbf{x}_r, \epsilon^l)) - \nabla \mathbb{E}_{q_\phi(\mathbf{z}_r|\mathbf{x}_r)} [\log q_\phi(\mathbf{z}_r|\mathbf{x}_r)] \right], \tag{10}$$

where $\epsilon^l$ are iid samples from $p(\epsilon)$. In practice, we evaluate the gradient in eq. 9 using a single sample from $p(\epsilon)$ ($L = 1$) and use ADADELTA for stochastic optimization [16].

**Choice of approximate posterior $q_\phi(\mathbf{z}|\mathbf{x})$:**   The AEVB approach to inference is appealing in its generality: it is well-defined for a large class of generative models $p_\theta(\mathbf{x}, \mathbf{z})$ and approximate posteriors $q_\phi(\mathbf{z}|\mathbf{x})$. In practice, however, the performance of the algorithm has a strong dependence upon the particular structure of these models. In our case, we use an approximate posterior that is designed explicitly to parameterize a temporally correlated approximate posterior [17]. We use a Gaussian approximate posterior,

$$q_\phi(\mathbf{z}_r|\mathbf{x}_r) = \mathcal{N}\left(\mu_\phi(\mathbf{x}_r), \Sigma_\phi(\mathbf{x}_r)\right), \tag{11}$$

where $\mu_\phi(\mathbf{x}_r)$ is a $mT \times 1$ mean vector and $\Sigma_\phi(\mathbf{x}_r)$ is a $mT \times mT$ covariance matrix. Both $\mu_\phi(\mathbf{x}_r)$ and $\Sigma_\phi(\mathbf{x}_r)$ are parameterized by observations $\mathbf{x}$ through a structured neural network, as described in detail in supplementary material. We can sample from this approximate by setting $p(\epsilon) \sim \mathcal{N}(0, I)$ and $h_\phi(\epsilon; \mathbf{x}) = \mu_\phi(\mathbf{x}) + \Sigma_\phi^{1/2}(\mathbf{x}_r)\epsilon$, where $\Sigma_\phi^{1/2}$ is the Cholesky decomposition of $\Sigma_\phi$.

This approach is similar to that of [8], except that we impose a block-tridiagonal structure upon the precision matrix $\Sigma_\phi^{-1}$ (rather than a diagonal covariance), which can express rich temporal correlations across time (essential for the posterior to capture the smooth, correlated trajectories typical of LDS posteriors), while remaining tractable with a computational complexity that scales linearly with $T$, the length of a trial.

## 5   Experiments

### 5.1   Simulation experiments

**Linear dynamical system models with shared, fixed rate function:**   Our AEVB approach in principle permits inference in any latent LDS model. To illustrate this flexibility, we simulate 3 datasets from previously-proposed models of neural responses. In our simulations, each data-generating model has a latent LDS state of $m = 2$ dimensions, as described by eq. 1 and eq. 2. In all data-generating models, spike rates depend on the latent state variable through a fixed link function $f$ that is common across neurons. Each data-generating model has a distinct observation model (eq. 3): Bernoulli (logistic link), Poisson (exponential link), or negative-binomial (exponential link).

We compare PLDS and GCLDS model fits to each datasets, using both our AEVB algorithm and two EM-based inference algorithms: LapEM (which approximates $p(\mathbf{z}|\mathbf{x})$ with a multivariate Gaussian by Laplace approximation in the E-step [1, 3]) and VBDual (which approximates $p(\mathbf{z}|\mathbf{x})$ with a multivariate Gaussian by variational inference, through optimization in the dual space [18, 3]). Additionally, we fit PfLDS and GCfLDS models with the AEVB algorithm. On this linear simulated data we do not expect these nonlinear techniques to outperform linear methods. In all simulation studies we generate 20 training trials and 20 testing trials, with 100 simulated neurons and 200 time bins for each trial. Results are averaged across 10 repeats.

We compare the predictive performance and running times of the algorithms in Table 1. For both PLDS and GCLDS, our AEVB algorithm gives results comparable to, though slightly worse than, the

Table 1: Simulation results with a linear observation model: Each column contains results for a distinct experiment, where the true data-generating distribution was either Bernoulli, Poisson or Negative-binomial. For each generative model and inference algorithm (one per row), we report the predictive log likelihood (PLL) and computation time (in minutes) of the model fit to each dataset. We report the PLL (divided by number of observations) on test data, using one-step-ahead prediction. When training a model using the AEVB algorithm, we run 500 epochs before stopping. For LapEM and VBDual, we initialize with nuclear norm minimization [2] and stop either after 200 iterations or when the ELBO (scaled by number of time bins) increases by less than $\epsilon = 10^{-9}$ after one iteration.

| Model | Inference | Bernoulli | | Poisson | | Negative-binomial | |
|---|---|---|---|---|---|---|---|
| | | PLL | Time | PLL | Time | PLL | Time |
| | LapEM | -0.446 | 3 | -0.385 | 5 | -0.359 | 5 |
| PLDS | VBDual | -0.446 | 157 | -0.385 | 170 | -0.359 | 138 |
| | AEVB | -0.445 | 50 | -0.387 | 55 | -0.363 | 53 |
| PfLDS | AEVB | -0.445 | 56 | -0.387 | 58 | -0.362 | 50 |
| | LapEM | -0.389 | 40 | -0.385 | 97 | -0.359 | 101 |
| GCLDS | VBDual | -0.389 | 131 | -0.385 | 126 | -0.359 | 127 |
| | AEVB | -0.390 | 69 | -0.386 | 75 | -0.361 | 73 |
| GCfLDS | AEVB | -0.390 | 72 | -0.386 | 76 | -0.361 | 68 |

LapEM and VBEM algorithms. Although PfLDS and GCfLDS assume a much more complicated generative model, both provide comparable predictive performance and running time. We note that while LapEM is competitive in running time in this relatively small-data setting, the AEVB algorithm may be more desirable in a large data setting, where it can learn model parameters even before seeing the full dataset. In constrast, both LapEM and VBDual require a full pass through the data in the E-step before the M-step parameter updates. The recognition model used by AEVB can also be used to initialize the LapEM and VBEM in the linear LDS cases.

**Simulation with "grid cell" type response:** A grid cell is a type of neuron that is activated when an animal occupies any vertex of a grid spanning the environment [19]. When an animal moves along a one-dimensional line in the space, grid cells exhibit oscillatory responses. Motivated by the response properties of grid cells, we simulated a population of 100 spiking neurons with oscillatory link functions and a shared, one-dimensional input $\mathbf{z}_{rt} \in \mathbb{R}$ given by,

$$\mathbf{z}_{r1} = 0, \tag{12}$$
$$\mathbf{z}_{r(t+1)} \sim \mathcal{N}(0.99\mathbf{z}_{rt}, 0.01). \tag{13}$$

The log firing rate of each neuron, indexed by $i$, is coupled to the latent variable $\mathbf{z}_{rt}$ through a sinusoid with a neuron-specific phase $\phi_i$ and frequency $\omega_i$

$$\mathbf{x}_{rti} \sim \text{Poisson}\left(\lambda_{rit} = \exp(2\sin(\omega_i \mathbf{z}_{rt} + \phi_i) - 2)\right). \tag{14}$$

We generated $\phi_i$ uniformly at random in the region $[0, 2\pi]$ and set $\omega_i = 1$ for neurons with index $i \leq 50$ and $\omega_i = 3$ for neurons with index $i > 50$. We simulated 150 training and 20 testing trials, each with $T = 120$ time bins. We repeated this simulated experiment 10 times.

We compare performance of PLDS with PfLDS, both with a 1-dimensional latent variable. As shown in Figure 1, PLDS is not able to adapt to the nonlinear and non-monotonic link function, and cannot recover the true latent variable (left panel and bottom right panel) or spike rate (upper right panel). On the other hand the PfLDS model captures the nonlinearity well, recovering the true latent trajectory. The one-step-ahead predictive log likelihood (PLL) on a held-out dataset for PLDS is -0.622 (se=0.006), for PfLDS is -0.581 (se=0.006). A paired t-test for PLL is significant ($p < 10^{-6}$).

## 5.2 Applications to experimentally-recorded neural data

We analyze two multi-neuron spike-train datasets, recorded from primary visual cortex and primary motor cortex of the macaque brain, respectively. We find that fLDS models outperform PLDS in terms of predictive performance on held out data. Further, we find that the latent trajectories uncovered by fLDS are lower-dimensional and more structured than those recovered by PLDS.

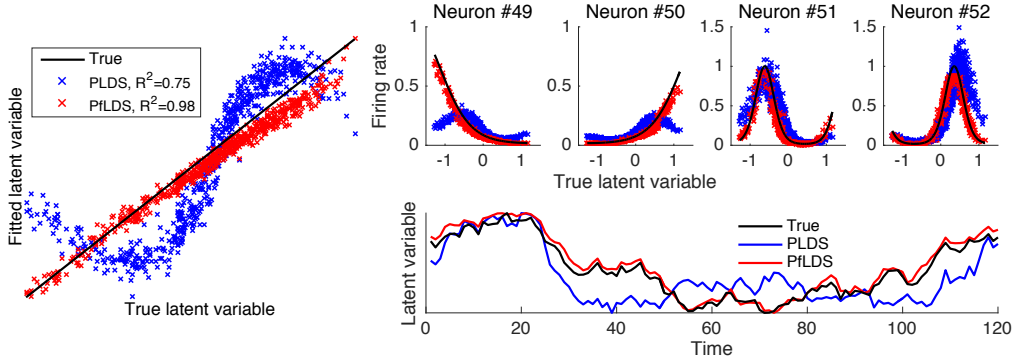

Figure 1: Sample simulation result with "grid cell" type response. *Left panel:* Fitted latent variable compared to true latent variable; *Upper right panel:* Fitted rate compared to the true rate for 4 sample neurons; *Bottom right panel:* Inferred trace of the latent variable compared to true latent trace. Note that the latent trajectory for a 1-dimensional latent variable is identifiable up to multiplicative constant, and here we scale the latent variables to lie between 0 and 1.

**Macaque V1 with drifting grating stimulus with single orientation:** The dataset consists of 148 neurons simultaneously recorded from the primary visual cortex (area V1) of an anesthetized macaque, as described in [20] (array 5). Data were recorded while the monkey watched a 1280ms movie of a sinusoidal grating drifting in one of 72 orientations: $(0°, 5°, 10°,...)$. Each of the 72 orientations was repeated $R = 50$ times. We analyze the spike activity from 300ms to 1200ms after stimulus onset. We discretize the data at $\Delta t = 10$ms, resulting in $T = 90$ timepoints per trial. Following [20], we consider the 63 neurons with well-behaved tuning-curves. We performed both single-orientation and whole-dataset analysis.

We first use 12 equal spaced grating orientations $(0°, 30°, 60°,...)$ and analyze each orientation separately. To increase sample size, for each orientation we pool data from the 2 neighboring orientations (e.g. for orientation $0°$, we include data from orientation $5°$ and $355°$), thereby getting 150 trials for each dataset (we find similar, but more variable, results when we do not include neighboring orientations). For each orientation, we divide the data into 120 training trials and 30 testing trials. For PfLDS we further divide the 120 training trials into 110 trials for fitting and 10 trials for validation (we use the ELBO on validation set to determine when to stop training). We do not include a stimulus model, but rather perform unsupervised learning to recover a low-dimensional representation that combines both internal and stimulus-driven dynamics.

We take orientation $0°$ as an example (the other orientations exhibit a similar pattern) and compare the fitted result of PLDS and PfLDS with a 2-dimensional latent space, which should in principle adequately capture the oscillatory pattern of the neural responses. We find that PfLDS is able to capture the nonlinear response charateristics of V1 complex cells (Fig. 2(a), black line), while PLDS can only reliably capture linear responses (Fig. 2(a), blue line). In Fig. 2(b)(c) we project all trajectories onto the 2-dimensional latent manifold described by the PfLDS. We find that both techniques recover a manifold that reveals the rotational structure of the data; however, by offsetting the nonlinear features of the data into the observation model, PfLDS recovers a much cleaner latent representation(Fig. 2(c)).

We assess the model fitting quality by one-step-ahead prediction on a held-out dataset; we compare both percentage mean squared error (MSE) reduction and negative predictive log likelihood (NLL) reduction. We find that PfLDS recovers more compact representations than the PLDS, for the same performance in MSE and NLL. We illustrate this in Fig. 2(d)(e), where PLDS requires approximately 10 latent dimensions to obtain the same predictive performance as an PfLDS with 3 latent dimensions. This result makes intuitive sense: during the stimulus-driven portion of the experiment, neural activity is driven primarily by a low-dimensional, oscillatory stimulus drive (the drifting grating). We find that the highly nonlinear generative models used by PfLDS lead to *lower*-dimensional and hence *more* interpretable latent-variable representations.

To compare the performance of PLDS and PfLDS on the whole dataset, we use 10 trials from each of the 72 grating orientations (720 trials in total) as a training set, and 1 trial from each orientation

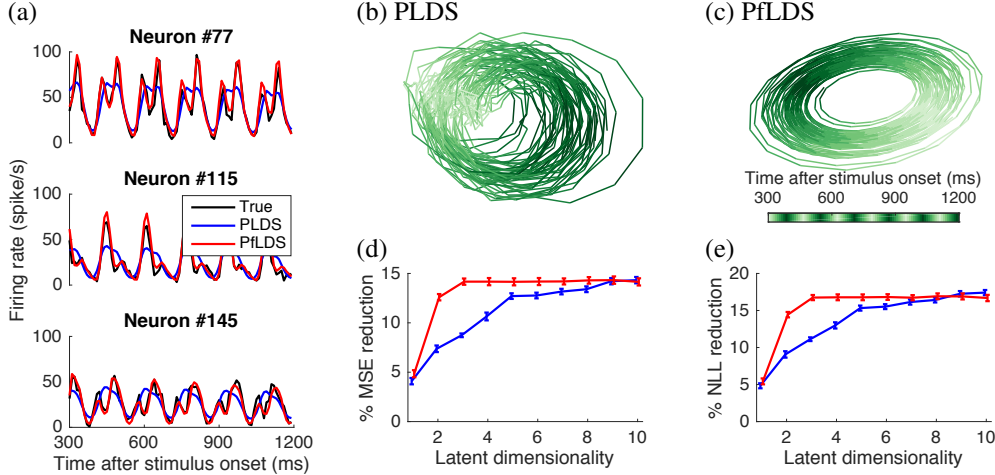

Figure 2: Results for fits to Macaque V1 data (single orientation) (a) Comparing true firing rate (black line) with fitted rate from PLDS (blue) and PfLDS (red) with 2 dimensional latent space for selected neurons (orientation $0°$, averaged across all 120 training trials); (b)(c) 2D latent-space embeddings of 10 sample training trials, color denotes phase of the grating stimulus (orientation $0°$); (d)(e) Predictive mean square error (MSE) and predictive negative log likelihood (NLL) reduction with one-step-ahead prediction, compared to a baseline model (homogeneous Poisson process). Results are averaged across 12 orientations.

as a test set. For PfLDS we further divide the 720 trials into 648 for fitting and 72 for validation. We observe in Fig. 3(a)(b) that PfLDS again provides much better predictive performance with a small number of latent dimensions. We also find that for PfLDS with 4 latent dimensions, when we projected the observation into the latent space and take the first 3 principal components, the trajectory forms a torus (Fig. 3(c)). Once again, this result has an intuitive appeal: just as the sinusoidal stimuli (for a fixed orientation, across time) are naturally embedded into a 2D ring, stimulus variation in orientation (at a fixed time) also has a natural circular symmetry. Taken together, the stimulus has a natural toroidal topology. We find that fLDS is capable of uncovering this latent structure, even without any prior knowledge of the stimulus structure.

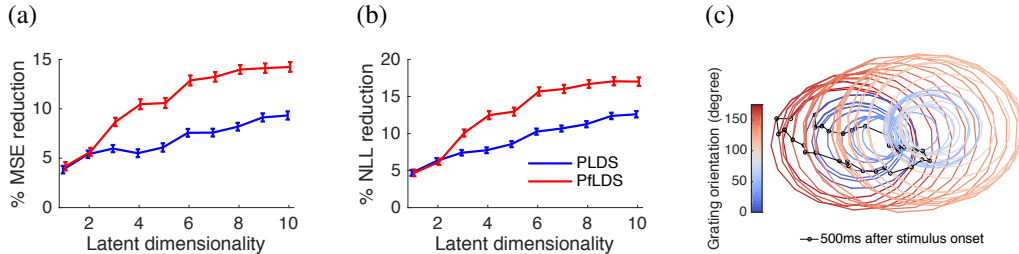

Figure 3: Macaque V1 data fitting result (full data) (a)(b) Predictive MSE and NLL reduction. (c) 3D embedding of the mean latent trajectory of the neuron activity during 300ms to 500ms after stimulus onset across grating orientations $0°, 5°, ..., 175°$, here we use PfLDS with 4 latent dimensions and then project the result on the first 3 principal components. A video for the 3D embedding can be found at `https://www.dropbox.com/s/cluev4fzfsob4q9/video_fLDS.mp4?dl=0`

**Macaque center-out reaching data:** We analyzed the neural population data recorded from the macaque motor cortex(`G20040123`), details of which can be found in [11, 1]. Briefly, the data consist of simultaneous recordings of 105 neurons for 56 cued reaches from the center of a screen to 14 peripheral targets. We analyze the reaching period (50ms before and 370ms after movement onset) for each trial. We discretize the data at $\Delta t = 20$ms, resulting in $T = 21$ timepoints per trial. For each target we use 50 training trials and 6 testing trials and fit all the 14 reaching targets together (making 700 training trials and 84 testing trials). We use both Poisson and GC noise models, as GC

has the flexibility to capture the noted under-dispersion of the data [3]. We compare both PLDS and PfLDS as well as GCLDS and GCfLDS fits. For both PfLDS and GCfLDS we further divide the training trials into 630 for fitting and 70 for validation.

As is shown in figure Fig. 4(d), PfLDS and GCfLDS with latent dimension 2 or 3 outperforms their linear counterparts with much larger latent dimensions. We also find that GCLDS and GCfLDS models give much better predictive likelihood than their Poisson counterparts. On figure Fig. 4(b)(c) we project the neural activities on the 2 dimensional latent space. We find that PfLDS (Fig. 4(c)) clearly separates the reaching trajectories and orders them in exact correspondence with the true spatial location of the targets.

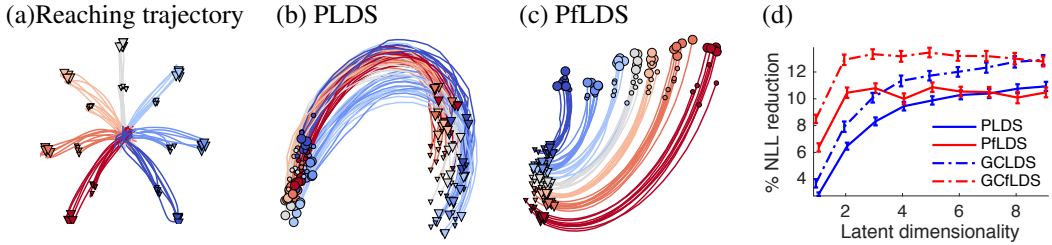

Figure 4: Macaque center-out reaching data analysis: (a) 5 sample reaching trajectory for each of the 14 target locations. Directions are coded by different color, and distances are coded by different marker size; (b)(c) 2D embeddings of neuron activity extracted by PLDS and PfLDS, circles represent 50ms before movement onset and triangles represent 340ms after movement onset. Here 5 training reaches for each target location are plotted; (d) Predictive negative log likelihood (NLL) reduction with one-step-ahead prediction.

# 6   Discussion and Conclusion

We have proposed fLDS, a modeling framework for high-dimensional neural population data that extends previous latent, low-dimensional linear dynamical system models with a flexible, nonlinear observation model. Additionally, we described an efficient variational inference algorithm suitable for fitting a broad class of LDS models – including several previously-proposed models. We illustrate in both simulation and application to real data that, even when a neural population is modulated by a low-dimensional linear dynamics, a latent variable model with a linear rate function fails to capture the true low-dimensional structure. In constrast, a fLDS can recover the low-dimensional structure, providing better predictive performance and more interpretable latent-variable representations.

[21] extends the linear Kalman filter by using neural network models to parameterize both the dynamic equation and the observation equation, they uses RNN based recognition model for inference. [22] composes graphical models with neural network observations and proposes structured auto encoder variational inference algorithm for inference. Ours focus on modeling count observations for neural spike train data, which is orthogonal to the papers mentioned above.

Our approach is distinct from related manifold learning methods [23, 24]. While most manifold learning techniques rely primarily on the notion of nearest neighbors, we exploit the temporal structure of the data by imposing strong prior assumption about the dynamics of our latent space. Further, in contrast to most manifold learning approaches, our approach includes an explicit generative model that lends itself naturally to inference and prediction, and allows for count-valued observations that account for the discrete nature of neural data.

Future work includes relaxing the latent linear dynamical system assumption to incorporate more flexible latent dynamics (for example, by using a Gaussian process prior [12] or by incorporating a nonlinear dynamical phase space [25]). We also anticipate our approach may be useful in applications to neural decoding and prosthetics: once trained, our approximate posterior may be evaluated in close to real-time.

A Python/Theano [26, 27] implementation of our algorithms is available at `http://github.com/earcher/vilds`.

## Footnotes

[2]Here, we consider a posterior $q_\phi(\mathbf{z}|\mathbf{x})$ that is conditioned explicitly upon $\mathbf{x}$. However, this is not necessary for variational inference.

[3]The approximate posterior is also sometimes called a "recognition model".

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
