[Supplementary Material · supp_fLDS.pdf]

# Supplementary material: *Linear dynamical neural population models through nonlinear embedding*

## Acknowledgments

Funding for this research was provided by the Sloan Foundation, the McKnight Foundation, and Simons Foundation Global Brain Research Awards 325233, 325171, 365002, ONR N00014-14-1-0243, ARO MURI W911NF-12-1-0594, DARPA N66001- 15-C-4032 (SIMPLEX), and a Google Faculty Research award; in addition, this work was supported by the Intelligence Advanced Research Projects Activity (IARPA) via Department of Interior/ Interior Business Center (DoI/IBC) contract number D16PC00003. The U.S. Government is authorized to reproduce and distribute reprints for Governmental purposes notwithstanding any copyright annotation thereon. Disclaimer: The views and conclusions contained herein are those of the authors and should not be interpreted as necessarily representing the official policies or endorsements, either expressed or implied, of IARPA, DoI/IBC, or the U.S. Government. Thanks to Arnulf Graf, Adam Kohn, Tony Movshon, and Mehrdad Jazayeri for providing the V1 data. Thanks to Krishna V. Shenoy, Byron Yu, Gopal Santhanam and Stephen Ryu for providing the motor cortical data.

## A    Temporally correlated approximate posterior

Below we detail the recognition model $q_\phi(\mathbf{z}|\mathbf{x})$ we used in our auto-encoder variational inference algorithm in fitting fLDS. For further details, see [17].

We construct $q_\phi(\mathbf{z}|\mathbf{x})$ as a product of factors across time,

$$q_\phi(\mathbf{z}_r|\mathbf{x}_r) \propto \prod_{t=1}^{T} q_\phi(\mathbf{z}_{rt}|\mathbf{z}_{r(t-1)})q_\phi(\mathbf{z}_{rt}|\mathbf{x}_{rt})q_\phi(\mathbf{z}_{r1}). \tag{15}$$

such that:

$$q_\phi(\mathbf{z}_{r1}) \sim \mathcal{N}(\tilde{\mu}_1, \tilde{Q}_1), \tag{16}$$

$$q_\phi(\mathbf{z}_{rt}|\mathbf{z}_{r(t-1)}) \sim \mathcal{N}(\tilde{A}\mathbf{z}_{r(t-1)}, \tilde{Q}), \tag{17}$$

$$q_\phi(\mathbf{z}_{rt}|\mathbf{x}_{rt}) \sim \mathcal{N}(m_{\tilde{\psi}}(\mathbf{x}_{rt}), c_{\tilde{\psi}}(\mathbf{x}_{rt})). \tag{18}$$

The parameters $\tilde{A}$, $\tilde{Q}$ and $\tilde{Q}_1$ are $m \times m$ matrices that control the smoothness of the posterior, and are analogous to the LDS parameters appearing in eq. 1 and eq. 2. Functions $m_{\tilde{\psi}}(\cdot) : \mathbb{R}^n \to \mathbb{R}^m$ and $c_{\tilde{\psi}}(\cdot) : \mathbb{R}^n \to \mathbb{R}^{m \times m}$ are nonlinear functions of observations $\mathbf{x}_t \in \mathbb{R}^n$, parameterized by $\tilde{\psi}$. To ensure non-negative definiteness of $c_{\tilde{\psi}}(\mathbf{x}_{rt})$, we first map the observations $\mathbf{x}_t$ to the square root of the precision matrix. We parameterize a matrix-valued function $r_{\tilde{\psi}}(\cdot) : \mathbb{R}^n \to \mathbb{R}^{m \times m}$ by a feed-forward neural network, and set $c_{\tilde{\psi}}(\mathbf{x}_{rt}) = \left( r_{\tilde{\psi}}(\mathbf{x}_{rt})r_{\tilde{\psi}}(\mathbf{x}_{rt})^T \right)^{-1}$. To summarize, the recognition model is parameterized by $\phi = (\tilde{\mu}_1, \tilde{A}, \tilde{Q}, \tilde{Q}_1, \tilde{\psi})$.

This product of Gaussian factors also has a Gaussian functional form, with block-tridiagonal inverse covariance. Normalizing recovers the multivariate Gaussian representation of eq. 11, where

$$\Sigma_\phi(\mathbf{x}_r) = \left( \mathbf{D}^{-1} + \mathbf{C}_\phi^{-1}(\mathbf{x}_r) \right)^{-1} \tag{19}$$

$$\mu_\phi(\mathbf{x}_r) = \left( \mathbf{D}^{-1} + \mathbf{C}_\phi^{-1}(\mathbf{x}_r) \right)^{-1} \mathbf{C}_\phi^{-1}(\mathbf{x}_r)\mathbf{M}_\phi(\mathbf{x}_r). \tag{20}$$

here $\mathbf{D} = (I - \mathbf{A})^{-\mathrm{T}}\mathbf{Q}(I - \mathbf{A})^{-1}$, where

$$\mathbf{Q} = \begin{bmatrix} \tilde{Q}_1 & & & \\ & \tilde{Q} & & \\ & & \ddots & \\ & & & \tilde{Q} \end{bmatrix}, \quad \mathbf{A} = \begin{bmatrix} 0 & & & & \\ \tilde{A} & 0 & & & \\ & \tilde{A} & 0 & & \\ & & \ddots & \ddots & \\ & & & \tilde{A} & 0 \end{bmatrix}, \quad (21)$$

and

$$\mathbf{C}_{\tilde{\psi}}(\mathbf{x}_r) = \begin{bmatrix} c_{\tilde{\psi}}(\mathbf{x}_{r1}) & & & \\ & c_{\tilde{\psi}}(\mathbf{x}_{r2}) & & \\ & & \ddots & \\ & & & c_{\tilde{\psi}}(\mathbf{x}_{rT}) \end{bmatrix}, \quad \mathbf{M}_{\tilde{\psi}}(\mathbf{x}) = \begin{bmatrix} m_{\tilde{\psi}}(\mathbf{x}_{r1}) \\ \vdots \\ m_{\tilde{\psi}}(\mathbf{x}_{rT}) \end{bmatrix} \in \mathbb{R}^{mT} \quad (22)$$

## B  Neural network structure for generative model and approximate posterior

In all our experiments with PfLDS and GCfLDS, we parameterize $f_\psi(\cdot) : \mathbb{R}^m \to \mathbb{R}^n$ using a feed-forward neural network with 2 hidden layers, each containing 60 nodes using tanh nonlinearity. For PfLDS we transform the final output layer by an exponential function to ensure the positivity of the rate.

For the approximate posterior described in section A, we parameterize $m_{\tilde{\psi}}(\cdot) : \mathbb{R}^n \to \mathbb{R}^m$ and $r_{\tilde{\psi}}(\cdot) : \mathbb{R}^n \to \mathbb{R}^{m \times m}$ by a neural network with two hidden layers, each containing 60 nodes using tanh nonlinearity. Here the hidden layers are shared for $m_{\tilde{\psi}}(\cdot)$ and $r_{\tilde{\psi}}(\cdot)$.

## C  Description of the video

We include a video (`https://www.dropbox.com/s/cluev4fzfsob4q9/video_fLDS.mp4?dl=0`) illustrating the latent-space projection of the macaque V1 data we analyzed in the paper. We fit the data with a PfLDS with $4$ latent dimensions, and plot the first 3 principal components of the inferred latent trajectory. We use data from 300ms to 1200ms after stimulus onset, and for each grating orientation we plot the mean trajectory of the 10 training trials used to fit the model. To illustrate the relationship between the latent space projection and the observed data, we show the latent trajectories for 3 single directions ($0°$, $60°$ and $120°$) alongside the spike rasters of one associated trial. We then show the latent trajectories for all 36 orientations, from $0°$ to $175°$ (results for directions from $180°$ to $355°$ are similar). We observe that the projection of neuron activity corresponding to each grating orientation forms a circle, agreeing well with the periodicity of the sinusoidal stimulus (temporal frequency 6.25Hz) and also the periodicity of the neural activity (as can be seen from the spike raster). The projection of the whole dataset forms a torus, which also agrees well with the stimulus structure.