[Reviews · NeurIPS 2016]

Reviewer 1

Summary

proposed novel nonlinear population model and corresponding inference algorithm.

Qualitative Assessment

the idea/paper is great, i'm a huge fan. i was, however, confused about a few things. below, i have some general comments, followed by more specifics. GENERAL 1. accuracy vs. sample size: in statistics, "efficiency" is key (accuracy as a function of sample size). i want to understand the efficiency for this method for both simulations as well as the real data. ie, please plot error vs sample size for all the methods in each scenario. 2. dimensionality vs. kernel: it is well know that in many cases complexity can be "pushed around", either by keeping more dimensions, or by adding a nonlinear kernel. i want to understand better how many dimensions the linear model needs to get the accuracy of the nonlinear one. ie, please plot error vs latent dimensionality for both simulations and real data using both your nonlinear stuff and the linear (inhomogeneous) stuff. 3. baselines: a great feature of simulations is that one can compute bayes optimal performance, to provide an "upper bound". we can also compute naive performance (eg, a homogeneous poisson model), which provide a null "lower bound". in the real data example, you plotted lower bound. in both simulation settings, for all experiments, please provide lower and upper bounds. 4. notation: i couldn't quite follow it. SPECIFICS 1. notation: i found it obtuse, perhaps there is no better way. but, for example, why is P a function of \ell instead of only \lambda, which is itself a function of \ell? also, why are \ell and f() separated? it is because f can be a function of each neuron, and \ell is not? and if \lambda = \ell, i'm not sure why \lambda exists too. it was also a bit unclear what role "i" plays: "the i^th element of a deterministic rate function"? but also i \in {1,...,n}. so i assumed i indexed neuron, so what does it mean to say it is the i^th element of a rate function? it indexed the square bracket, which i guess means that it indexed \ell? so, i think i would have written: x_rti | z_rt ~ P( f_i(z_rt)) in any case, please clarify your choices or greatly simplify notation. if the obtuseness is useful for connecting to other stuff, maybe introduce it as simple as possible first, and then connect it to the other ones? 2. i also didn't understand "we represent f_\psi through a parametric neural network model" what is a "nonparametric" neural network model, such that you are distinguishing? maybe the word "parametric" in not necessary? please remove or clarify 3. \theta = (...). i was a bit confused which of those parameters index a neuron, and which index the whole population. the dimensionality of \theta was also a bit unclear to me, i realize it is a function of m & n (and the neural network). but what happened to say, c & d in the PLDS model? are they "eaten" by \psi, meaning you don't even represent them? please clarify which parameters are neuron specific, and the dimensionality of \theta (possibly different dimensionalities for the different special cases). 4. i believe that PfLDS is a generalization of fLDS, insofar as PfLDS has a nonlinear function of z, and fLDS has a linear one. in other words, the rates are: exp[ c*z+d ] for fLDS, and exp[ f_\psi(x) ] for PfLDS. is that correct? and the same kind of thing for GCfLDS? please clarify the text around lines 104-107. 5. "we append the noise model to the acroynm" i think you mean "prepend"? "append" means put on the end. and acronym is spelled wrong. 6. table 1: i was a bit confused about a few things here. A) are these numbers good? they are simulations, so you could use the true parameters and estimate the upper bound, and use some null/dumb model (like constant rate) to provide a lower bound. i just have no intuition for what PLL or time should be in any of these cases. B) assuming my comment #4 above is correct, then the fact that PfLDS does nearly identical to fLDS and the same is true for GCfLDS, suggests that f_\psi(.) might have learned a *linear* function (ie, gotten it correct). you don't show it, though you do show something similar for the next simulations. please show the estimated rate functions (maybe in appendix if you don't have room). C) you mention the number of trials, neurons, and time bins, as well as the architecture of f_\psi, but not the firing rates or counts. the estimation error is closely linked to spike count, ie, if your network only fired very sparsely, all estimates would be poor, and your nonlinear estimator would be even worse than the linear ones in many cases. so, at a minimum, please clarify some summary statistics for the number of spikes per time (maybe per neuron, or something like that). much better: please conduct an experiment where you compute accuracy as a function of # of spikes in training trials (or some other appropriate measure). i would expect that the linear estimators outperform the nonlinear ones until spike count is sufficiently high, but i have no intuition whatsoever in terms of how high it must be, in particular under various model misspecification scenarios that you have already described. the same experiment could be applied to the grid cell simulation. 7. "A paired t-test for PLL is significant (p < 106)." a t-test? is PLL really gaussian? i would have thought a wilcoxon test would be much more meaningful. 8. a major claim of the abstract and paper is: "We also show in application to two neural datasets that, compared to state-of-the-art neural population models, fLDS captures a much larger proportion of neural variability with a small number of latent dimensions" the results you provide indicate that for sufficiently large sample size, upon conditioning on latent dimensionality, your method wins. this is not particularly surprising, because yours is explicitly a nonlinear generalization of the others, so given enough data, it should. but, this improvement comes at a cost (2 costs). a) it takes more time to fit b) the understanding of each neuron now requires a nonlinear function, rather than a linear one. a natural question i have therefore is: how does performance change with the number of latent dimensions. in other words, above i asked for experiments where you conditioned on latent dimension, and change sample size. i'd also like to see experiments where you condition on sample size, and change the number of latent dimensions. for example, if adding a single latent dimension to the linear models recovers the additional variance that you obtain from your nonlinear one, it is not obvious which approach would be preferred. i suspect that this will not be the case, nonetheless, experimental demonstration would be nice. 9. "cleaner" and "more structured" are words you used to compare the estimated latent states. these are a bit informal. i wonder if you can quantify those concepts in a meaningful way? 10. "negative predictive log likelihood" why did you switch from PLL to NLL? seems like staying with one or the other throughout would be advantageous. 11. "We find 229 that the highly nonlinear generative models used by PfLDS lead to lower-dimensional and hence 230 more interpretable latent-variable representations." is 1D + nonlinear more interpretable than 2D+linear? i think this is debatable. in particular, the highly nonlinear generative models estimated by PfLDS have a highly nonlinear estimate amid the representation, which may or may not help interpretability. the latent representations are perhaps more interpretable. this is related to my previous comment. 12. Figure 2: why do (d) and (e) compare to a *homogeneous* Poisson process, rather than PLDS? i think adding PLDS to this figure would satisfy my desire for understanding the tradeoff between dimensions and nonlinearity in real data (though not in simulation). 13. how should one choose the dimensionality (and neural network architecture) when using your approach? presumably, both depend on sample size. some guidance would be appreciated. - jv

Confidence in this Review

3-Expert (read the paper in detail, know the area, quite certain of my opinion)


Reviewer 2

Summary

The authors propose a method for extracting low-dimensional neural trajectories from neural spike count data using a hierarchical model which assumes linear latent dynamics and observation models which allow for a non-linear relationships between latent variables and a rate parameter. AnNeural network provides the non-linearity between latents and and a rate parameter for each neuron. The authors propose auto-encoding variation Bayes (AEVB) as a means of simultaneously learning model parameters and performing inferences. Simulated results demonstrate that the proposed methods are nearly comparable to existing linear methods when data is simulated without a non-linearity and they are able to better model data with simulated non-linearities. Results on real V1 and M1 data show that the proposed methods are better able to model the observed data with a latent state space of reduced dimensionality compared to existing methods.

Qualitative Assessment

Methods for extracting low-dimensional representations of neural data are commonly used in neuroscience and the development of better methods for doing so has been the subject of much research in the computational community. While the value of incorporating observation models which allow for non-linear relationships between latent state and observed counts has been generally acknowledged, doing so in a manner that maintains computational tractability is challenging. The proposed work is therefore notable as it presents methods which are computationally tractable and produce impressive results on real data. The ability of the proposed methods to fit data with a smaller number of latents than competing methods is particularly notable as it may allow for representations of data which are easier to visualize, thereby making it easier to recognize important aspects of a data set. With this said, the clarity of the paper, while overall very good, could be improved in some parts. The description of AEVB Is very brief. In particular, it is not clear how the function g_phi(x, \epsilon) is chosen. Perhaps more important from the point of interpretation is that it appears there is a single neural network that links between latent state and the firing rate parameter for all neurons. This seems to discard the conditional independence assumption (that neurons are independent conditioned on state) that is common to most existing methods. It seems that this need not be problematic (perhaps even beneficial) - however it does change the interpretation of the latents. In models with conditionally independent neurons, it seems reasonable to think of the latent state as capturing a low-dimensional representation that explains the shared covariance structure of the modeled neural population. If the neural network linking latent state to firing rate is now able to also model the covariability of neurons, it seems there could be questions of basic identifiability and model selection. On the same note, the authors claim that the improved latent representations are due to the non-linear observation model, and it seems that improvements may be due to both the non-linearity as well as the relaxation of the conditional independence assumption. Discussion of these points would be helpful.

Confidence in this Review

2-Confident (read it all; understood it all reasonably well)


Reviewer 3

Summary

In this work, the authors introduce an updated method for extracting low-dimensional dynamics in neuronal population recordings. Their technique (PfLDS) provides more flexibility by allowing individual units to have individual (smooth) nonlinear rate functions. The results are both theoretically appealing, as the authors do a good job of explaining the intuitive appeal of their approach and the introduced technique provides a measured increase over previous work (PLDS).

Qualitative Assessment

The paper is clearly written, with the method clearly detailed (both mathematically and intuitively). My first major critique is that, while the method is tested on simulated and experimental data with known low-dimensional properties, it is not demonstrated how this method can be used to discover new, previously unknown properties of the neural response. I understand that the authors may not want to add additional data to the submission, in order to focus on introduction of the method; in that case, however, they could just as well add an additional, more exotic test form of the neural response that would serve to demonstrate how such a discovery may take place. Second, from a methodological standpoint, the authors do not adequately detail under what cases their method will fail. line 234: "We observe in Fig. 3(a)(b) that PfLDS again provides much better predictive performance with a small number of latent dimensions." Is this true? While this is clear in Figure 2, from my inspection of Figure 3ab, it appears that for 1-2 latent dimensions the two methods have the exact same performance. For increasing number of dimensions, however, the distinction between the two methods becomes clear. The authors should take care to describe their results with precision on this important point. line 255: Typo

Confidence in this Review

3-Expert (read the paper in detail, know the area, quite certain of my opinion)


Reviewer 4

Summary

In this manuscript, the authors developed a novel nonlinear generative model of neural activity. The model can deal with a supra- or sub- Poissonian observations using a generalized count distribution, and explain these observations by nonlineary linking it with latent dynamics. The latter nonlinearity allowed them to estimate even lower-dimensional structure than what the conventional linear models can find. Although simpler conventional models may better perform on simpler data, they demonstrate utilities of their methods using more realistic simulated examples as well as by real data analysis.

Qualitative Assessment

The manuscript is well written, and the method should clearly advance the field. I have only a few minor suggestions (see below), and generally recommend this manuscript for publication. Minor points of concerns: * Definition of fLDS is completely missing in the manuscript. What does 'f' stands for? * Since the improved sampling approach is still more complex than Laplace approximation, it is quite reasonable that the proposed method may not perform better when applied to simpler data or data with small sample size (Table 1). It is then recommended that the authors provide a guideline about which method should be used according to the data at hand by, for example, using information criteria such as Akaike Bayesian information criterion that combines a marginal likelihood with penalty in dimensionality of the hyper-parameters. * Eq. 11, 12, 13. Here, why do authors use x for latent processes, and y for observation while they use respectively z and x in the introduction of their model?

Confidence in this Review

2-Confident (read it all; understood it all reasonably well)


Reviewer 5

Summary

The authors consider the problem of modeling the time dependence of neural populations using latent variable dynamical models. While they assume that the dynamics of the latent variables are linear, and that the neurons are independent given the latent state, they allow for a very rich mapping from the latent state into the probabilities of single neuron responses. This is an important generalization over previously considered models which often impose strong modeling assumptions on the mapping from latent to observed space. They also provide an algorithm for fitting such models and demonstrate their usefulness on both artificial and real data.

Qualitative Assessment

I really enjoyed the paper. It is very important for neuroscience to look beyond simple linear-nonlinear models and take advantage of more powerful methods (e.g. neural networks) as done in this paper. Comments (ordered by importance): 1. Fig. 4b,c: Why do you show the latent trajectories for the Poisson models rather than for the general count models which clearly perform much better? It would be useful to have these at least in the supplement. I especially wonder whether the latent trajectories would be more separated when using GCLDS. 2. Fig. 3: Again, since you're doing additional dimensionality reduction using PCA, why don't you look at the model with 10 latents which is clearly much better than the model with only 4 latents (according to Fig. 3a,b)? 3. Line 223: The responses are discrete so it is not clear what is meant by MSE. Is it MSE between observed and predicted firing rates? Averaged over all neurons? 4. Line 125-126: I couldn't find what function g you used. Maybe it is something standard but for people who never used AEVB it is not obvious. 5. Line 179: You already used x to denote neuron responses so don't use it here as the latent variable. 6. Line 137: nT -> mT?

Confidence in this Review

2-Confident (read it all; understood it all reasonably well)


Reviewer 6

Summary

Following a long string of LDS models to perform dimensionality reduction on neural spike trains, this paper allows for nonlinear link functions between latent variables and observed spike counts. This is an important step, because presumably neural activity lies along a nonlinear, low-d manifold that can be better described with nonlinear methods than linear methods. One of the challenges employing nonlinear dimensionality reduction on spike trains is that neurons are incredibly noisy, and nonlinear dim reduction methods do not handle noise well. However, by making use of time information, one could hope to overcome such noise. This paper makes several strides, including making use of time information (with a linear dynamical system), allowing for nonlinear interactions between observations and latents, and providing a tractable fitting procedure. They provide experiments to test their method with both synthetic and experimental data.

Qualitative Assessment

I enjoyed this paper, as it will interest both machine learners and neuroscientists. It makes notable progress over other current LDS methods because it allows for nonlinear interactions between latent and observed spike count. I also found the method to be well tested---it provides better reconstruction performance than a state-of-the-art linear method, and seems to empirically work well with experimental data. I think this should be accepted to NIPS. My major comments are mainly interpretative. You should read the following comments as excitement for the paper! However, the community should decide how useful this method is for real neural data sets, not a reviewer. I am glad they plan to release code for this method. 1. For Figure 2, give intuition of why 2b is more "jagged" than 2c. Does the neural network function become more "jagged" or is it because of the approximate posterior that incorporates more smoothing? I imagine you can also increase a smoothing factor for PLDS? 2. If one applies PCA to the PSTHs, they will also see circular dynamics that correspond to the phase of the grating with the top two PCs. However, one could represent phase with just one latent variable. Thus, it seems PfLDS is not giving us much here in terms of understanding the dynamics of the population activity. Also, it's unclear how adaptation will show up in the latent space (I see you clipped trials off at 300-500ms to avoid this). 3. It's incredible how non-variable the latents look. There is a great deal of trial-to-trial variability in V1 responses, and a lot is shared across neurons. This means it can often be difficult to discern what a PSTH would look like by studying a single-trial neural trajectory. So it seems PfLDS can be beneficial to studying the stimulus dynamics (which also follow a linear dynamical system) on single trials. However, if one wants to study trial-to-trial variability, PfLDS does not seem to be an appropriate method. 4. Fig. 2d and 2e show that two latent variables capture about double the % than just one latent variable (in 2d, 5% for 1 latent vs. ~13% for two latents). This suggests that the second latent variable includes more information than the first latent variable. Thus, PfLDS does not order the dimensions in any way. This could be troublesome, if one has to assess the number of dimensions to include. Do you suggest applying PCA to the latents first, and choose the dimensionality that way? Including a way to assess dimensionality will be helpful. Fig. 3a and 3b also suggest the dimensions are not ordered. 5. Fig. 3c shows a torus, and makes intuitive sense. But that's not how the neural activity is structured in firing rate space. For example, if you summed up the spike counts across time for each trial, one could not decode orientation from Fig. 3c---Each trajectory would collapse to one point, and lie along the major axis of the torus. Thus, one could not discriminate between, say, 45 deg and 135 deg (they would be projected to similar points along the axis). However, in firing rate space, orientation is easily decoded (shown by the Graf et. al, 2009 paper), as one can see neural states for each orientation placed along a circle. Thus, it seems the PfLDS method is altering the structure of firing rate space, due to the constraints. This may be a good thing or a bad thing---probably depends on the problem. 6. Fig. 4c...similar comment as 5. For 2 latents in firing rate space, I would see the neural trajectories reach out along a circle (see Santhanam et al., 2009, about FA methods). I suppose we do see the rotational dynamics as seen in Churchland et al, 2012, right? 7. I am really worried about time....if a method takes over an hour to find latents for population activity in response to gratings, that's not good. You have crunch times listed for 40 trials, 100 neurons, and 200 time bins per trial---and it looks like it takes about an hour. I would suggest trying to improve the running time as much as possible before releasing the code. ---- responses to ratings Technical quality - I found the paper to be of sound technical content and analyses to show that their method is state-of-the-art. Novelty/originality - They presented new novelties, including an approximate prior and an approach to include nonlinearity. Potential impact: - The paper will be of interest to both the machine learning and neuroscience communities. The paper presents a nice problem about variational inference (and comparing different methods), while also showing the method works for real data. Clarity and presentation - The paper was well-organized, figures were clear and concise. I also enjoyed the bomber supplemental video. ---- minor comments - line 30, "subspace" --> "manifold", just seems more appropriate for an ML audience - line 73 (eqn. 2), it took a while to parse the equal signs inside an argument. I would suggest just having: P(lambda); lambda = .... - line 180, "log firing rate" --- could just say link function is exponential, I was wondering why you were taking the log of a firing rate - line 190, it would be nice to include how long your method took to run - line 199, I think you can better and more concisely give the experimental information. Since none of your analyses seem to use all 72 orientations, just say 36 orientations were presented. Likewise, only say you look at 63 neurons (no need to mention 148 neurons). - line 206, may want to include that V1 neurons have lots of trial-to-trial variability and reference that - line 221, I want more intuition about why PfLDS recovers a much cleaner latent. - line 265, there should be an "and" somewhere in that sentence

Confidence in this Review

3-Expert (read the paper in detail, know the area, quite certain of my opinion)